# Degradable Poly(ether-ester-urethane)s Based on Well-Defined Aliphatic Diurethane Diisocyanate with Excellent Shape Recovery Properties at Body Temperature for Biomedical Application

**DOI:** 10.3390/polym11061002

**Published:** 2019-06-05

**Authors:** Minghui Xiao, Na Zhang, Jie Zhuang, Yuchen Sun, Fang Ren, Wenwen Zhang, Zhaosheng Hou

**Affiliations:** 1College of Chemistry, Chemical Engineering and Materials Science, Shandong Normal University, Jinan 250014, China; xiaominghui98@163.com (M.X.); sdzxnzkay@163.com (N.Z.); yuchensun12365@163.com (Y.S.); 2Shandong Academy of Pharmaceutical Sciences, Shandong Provincial Key Laboratory of Biomedical Polymer, Jinan 250101, China; cpfzhuangjie@126.com; 3Success Bio-tech Co., Ltd., Jinan 250101, China; fangren_2008@126.com (F.R.); 15662698340@163.com (W.Z.)

**Keywords:** poly(ether-ester-urethane)s, poly(ethylene glycol), well-defined hard segments, degradability, shape memory behavior

## Abstract

The aim of this study is to offer a new class of degradable shape-memory poly(ether-ester-urethane)s (SMPEEUs) based on poly(ether-ester) (PECL) and well-defined aliphatic diurethane diisocyanate (HBH) for further biomedical application. The prepolymers of PECLs were synthesized through bulk ring-opening polymerization using ε-caprolactone as the monomer and poly(ethylene glycol) as the initiator. By chain extension of PECL with HBH, SMPEEUs with varying PEG content were prepared. The chemical structures of the prepolymers and products were characterized by GPC, ^1^H NMR, and FT-IR, and the effect of PEG content on the physicochemical properties (especially the shape recovery properties) of SMPEEUs was studied. The microsphase-separated structures of the SMPEEUs were demonstrated by DSC and XRD. The SMPEEU films exhibited good tensile properties with the strain at a break of 483%–956% and an ultimate stress of 23.1–9.0 MPa. Hydrolytic degradation in vitro studies indicated that the time of the SMPEEU films becoming fragments was 4–12 weeks and the introduction of PEG facilitates the degradation rate of the films. The shape memory properties studies found that SMPEEU films with a PEG content of 23.4 wt % displayed excellent recovery properties with a recovery ratio of 99.8% and a recovery time of 3.9 s at body temperature. In addition, the relative growth rates of the SMPEEU films were greater than 75% after incubation for 72 h, indicating good cytocompatibility in vitro. The SMPEEUs, which possess not only satisfactory tensile properties, degradability, nontoxic degradation products, and cytocompatibility, but also excellent shape recovery properties at body temperature, promised to be an excellent candidate for medical device applications.

## 1. Introduction

Shape-memory polymers (SMPs), as a kind of smart polymeric material, have the ability to remember and recover their permanent shape upon application of an external stimulus, such as heat, humidity, pH, light, electromagnetic induction, or a solvent [1,2,3,4,5,6]. Compared with shape-memory alloys (SMAs), SMPs have a lot of advantages, such as flexible transition temperatures, high recoverable strains, low density, low manufacturing cost, and easy processing [2]. Nowadays, SMPs have been proposed for application in several kinds of medical devices [7,8,9]. As an example, SMPs were used to fabricate medical bandages by the Luo group, and they found that SMPs could produce gradient pressures acting on an ulcer of the leg to accelerate blood circulation and the healing process [10]. Although SMPs are very useful, their applications in medical implant materials are limited because of their nonbiodegradability and low biocompatibility. For medical implantations, SMPs are required to be degradable, possess high biocompatibility and have a recovery temperature near the human body temperature. Therefore, it is urgently required to develop biocompatible and degradable SMPs with adequate tensile properties to fabricate novel kinds of medical devices [11,12,13]. In 2017, the Becker group [14] provided an overview of SMPs being used in medical applications, and after that, a large number of works related to biocompatible SMPs have been published [15,16,17].

Segmented polyurethanes (PUs), as one of the important types of shape-memory materials, have received attention recently because of their unique properties, such as high shape recoverability, a wide range of shape recovery temperatures, good tensile properties, and adequate biocompatibility [18,19]. The unique properties of segmented PUs are attributed to microphase separation, and the primary driving forces behind microphase separation are both the immiscibility of the soft and hard segments and the strong intermolecular interaction of H-bonding among the hard segments, with hydrogen bonding energies of 12~36 kJ/mol [20,21].

As implant medical devices, many commercial medical PUs, including Elasthane^TM^, Biomer^®^, Biospan^TM^, and ChronoFlex^®^ AR, are quintessentially synthesized based on 4,4’-diphenylmethane diisocyanates (MDI) [22,23]. Aromatic diamines are carcinogenic degradation products that are produced and released during the degradation process [24]. Clearly, PUs prepared from aliphatic diisocyanate can conquer these shortcomings; however, compared with MDI-based Pus, these types of PUs possess weak tensile properties because they lack a significant length of hard segments [25]. Long well-defined hard segments are significant for good tensile properties [26]. In our previous report [27], a kind of degradable medical PU containing well-defined hard segments was prepared by using the chain extender of aliphatic diurethane diisocyanate. The well-defined chemical structure of the hard segments enhances the microphase separation degree of the hard and soft segments, and in addition, the existence of multiple H-bonds between the urethane units gives a compact network structure physical-linked by H-bonds, which leads to comparative or even better tensile properties than MDI-based PUs.

On the other hand, most of the commercially available shape-memory PUs (SMPUs) are thermo-responsive. They have a higher recovery temperature (50–90 °C) than human body temperature, which makes them not particularly suitable for medical device applications as they would need high activation temperatures [28]. Therefore, it is desirable to design degradable SMPUs with a near-body recovery temperature for medical applications. Poly(ethylene glycol) (PEG) not only possesses good hydrophilicity, biocompatibility, and nontoxicity, but also exhibits excellent flexibility and a low glass-transition temperature [29,30]. The introduction of a PEG chain into hydrophobic polyester chains as the soft segment can produce a low transition temperature. Thus, the corresponding SMPUs based on the poly(ether-ester) have both biodegradability and a near-body recovery temperature. Besides, the flexible PEG chain can serve as a plasticizer to enhance the distance between the soft segments, and thus enhances the shape memory properties of the SMPUs, as other papers have reported [10,31,32].

This work describes the preparation and properties of a new class of shape-memory poly(ether-ester-urethane)s (SMPEEUs) based on poly(ether-ester) and well-defined aliphatic diurethane diisocyanate. The SMPEEUs are expected to have not only excellent shape recovery properties at body temperature, but also acceptable physicochemical properties, degradability, biocompatibility, and nontoxic degradation products for further biomedical application. First, the poly(ether-ester) of triblock poly(ε-caprolactone)-poly(ethylene glycol)-poly(ε-caprolactone) (PECL) was synthesized by bulk ring-opening polymerization with ε-caprolactone (ε-CL) as the monomer and PEG as the initiator. Then, the SMPEEUs with well-defined hard segments were obtained by one-step chain extension of PECL with diurethane diisocyanate. The chemical structures of the PECLs and SMPEEUs were characterized, and the influences of PEG content on the tensile properties, in vitro degradability, and shape-memory properties of the SMPEEU films were researched. In addition, the biocompatibility (cytocompatibility) of the SMPEEU films was evaluated by a cytotoxicity test.

## 2. Materials and Methods

### 2.1. Materials

*ε*-CL was purchased from Acros Chimica, Geel, Belgium and distilled from CaH_2_ under reduced pressure. Dibutyltin dilaurate (DBTDL) and 1,6-hexanediisocyanate (HDI) were supplied by Sigma-Aldrich Chemical Co., St. Louis, MO, USA and used without further purification. PEG (*M*_n_ = 400, 600, 1000, 2000 g/mol) and 1,4-butanediol (BDO) (Shanghai Aladdin Reagent Co., Shanghai, China) were dried at 110 °C under reduced pressure for about 4 h prior to use. Diurethane diisocyanate of 1,6-hexanediisocyanate-1,4-butanediol-1,6-hexanediisocyanate (HBH) was synthesized in our lab, as shown in Figure 1a according to our previous report [33] and HR-MS and NMR were adopted to confirm its chemical structure. *N*,*N*-dimethylformamide (DMF, Beijing Chemical Reagent Co., Ltd, Beijing, China) was dried with phosphorus pentoxide for 8 h and then distilled under vacuum before use. Other reagents (AR grade) were purified by standard methods.

### 2.2. Polyurethane Synthesis

#### 2.2.1. Synthesis of PECLs

OH-terminated PECLs were prepared by bulk ring-opening polymerization with ε-CL as the monomer and PEG as the initiator [34]. Briefly, the predetermined amount of ε-CL, PEG and catalyst DBTDL (0.25 wt % of ε-CL) were mixed in a vacuum flask at room temperature, and the system was deoxygenated with dry argon three times. After the flask was sealed, the reaction was performed at 140 °C under vacuum conditions (~50 Pa) with an oil bath for 24 h. A small amount of chloroform was added to the system to dissolve the raw product, and the solution was precipitated with cold diethyl ether to obtain the PECLs, which were then dried thoroughly under reduced pressure (~150 Pa) at room temperature. The PECLs based on PEG-400, PEG-600, PEG-1000, and PEG-2000, are named PECL-I, PECL-II, PECL-III, and PECL-IV, respectively. The reaction scheme is shown in Figure 1b, and the feed ratios and molecular weights of the PECLs are listed in Table 1.

#### 2.2.2. Preparation of SMPEEUs and SMPEEU Films

A representative process was as follows: A certain amount of PECL was added to a two-necked flask and the system was heated to 80 °C with an oil bath under dried argon. Then the DMF solution of HBH (~0.25 g/mL) was added dropwise into the system with mechanical stirring, and the ratio of –NCO/–OH was 1.02 (mol:mol). When the reaction became too viscous to be stirred, a small amount of DMF could be added to keep the reaction homogeneous. After that, the reaction mixture proceeded at the same temperature with dried argon until the characteristic peak of –NCO at 2250~2280 cm^−1^ disappeared completely in FT-IR (approximately 3.5 h), and then was diluted to ~0.045 g/mL with DMF. The diluted solution was degassed under reduced pressure and then gently poured into a polytetrafluoroethylene mold, in which most of the DMF was volatilized at 40 °C for five days. The last trace of DMF was removed under vacuum for two days to give semitransparent films with a thickness of 0.3 ± 0.02 mm. The SMPEEU films, based on PECL-I, PECL-II, PECL-III, and PECL-IV are named SMPEEU-I, SMPEEU-II, SMPEEU-III, and SMPEEU-IV, respectively. The reaction scheme is displayed in Figure 1c, and the molecular weight and feed ratios of the SMPEEUs are shown in Table 2.

### 2.3. Instruments and Characterization

Characterization: Nuclear magnetic resonance (NMR) spectra were obtained by employing a Bruker 400 MHz Avance II spectrometer (Rheinstetten, Germany) at room temperature. DMSO-d^6^ and CDCl_3_ were used as the solvents for HBH and polymer analysis, respectively. Fourier transform infrared (FT-IR) spectra were conducted on a Bruker Alpha spectrometer (Rheinstetten, Germany) in the range of 4000–400 cm^−1^_,_ with a resolution of 4 cm^−1^. The number average molecular weight (*M*_n_) and molecular weight dispersity (*Ð*_M_) were determined using Viscotec TriSEC302 (Kennesaw, GA, USA) gel permeation chromatography (GPC) at 35 °C, with tetrahydrofuran as the continuous phase (flow rate: 1.0 mL/min) and monodisperse polystyrene as the calibration standard.

Thermal transition: Thermal transition behaviors were researched with a Q20 differential scanning calorimeter (DSC) (TA Instrument, New Castle, DE, USA) under nitrogen (flow rate: 30 mL/min). Samples, which were encapsulated in standard aluminum pans with lids, were first heated up to 150 °C with a heating rate of 10 °C/min to relieve the thermal history. The samples were then cooled to −75 °C at 5 °C/min, and finally scanned from −75 to 150 °C at 10 °C/min. The reported thermal transition values were collected from the second heating cycle, and the value of the glass transition temperature (*T*_g_) was obtained from the center of the change of slope.

Crystallization behaviors: X-ray powder diffraction (XRD) was adopted to investigate the crystallization behaviors of SMPEEU films. The data were collected on a Max 2200PC power X-ray diffractometer (Rigaku, Tokyo, Japan) with Cu K_α_ radiation (wavelength: 1.54051 Å) at 40 kV and 20 mA. The sample holder containing film samples was scanned from 5° to 55° with a step size of 2*θ* = 0.02°.

Tensile properties: The tensile properties were tested with a single-column tensile test machine (Model HY939C, Dongguan Hengyu Instruments, Ltd., Dongguan, China) according to national standard GB/T1040.2-2006. The film samples were cut into dumb-bell shapes with a neck length and width of 4.0 and 30 mm, respectively. The tests were performed at room temperature with a cross-head speed of 50 mm/min. The ultimate stress, strain at break, and initial modulus were extracted from the stress-strain curves. Property values obtained here were the average of at least three tensile samples.

Bulk hydrophilicity: The water absorption was adopted to measure the bulk hydrophilicity of the SMPEEU films. A film disc with a 10.0 mm diameter was immersed in distilled water (10 mL) at 37 ± 0.1 °C until water absorption equilibrium was reached. The water absorption was obtained from the formula (1):(1)$${\mathrm{Water}}_{}\mathrm{absorption}\;(\%)=\frac{W_{s}-W_{o}}{W_{o}}\times 100$$
where *W*_s_ and *W*_o_ are the weights of the swollen and original samples, respectively. The results were the average of three samples.

Surface hydrophilicity: The film surface hydrophilicity was evaluated by sessile static water contact angle measurement, which was executed on an optical goniometer (CAM200, KSV Instruments, Helsinki, Finland) with a sessile drop method. Ultrapure water (~0.02 mL) was used as a liquid probe, and the measurement of each contact angle were finished within 5 s after steady state for the angle was reached. All the measurements were carried out at room temperature and the results were the average of at least six replicate measurements on each sample.

In vitro degradation: In vitro hydrolytic degradation tests of swollen films were carried out in phosphate-buffered saline (PBS) with a pH value of 7.4, employing the mass loss. The swollen film discs (diameter: 10.0 mm) were put into individual sealed vial containing PBS (10 mL), and incubated in a biochemical incubator at a temperature of 37 ± 0.1 °C. At given time intervals, the discs were taken out, wiped with filter paper, and weighed to determine the weight loss. The measurements were performed until the discs lost their tensile properties and became fragments. The weight loss was calculated according to the formula (2):(2)$${\mathrm{Weight}}_{}\mathrm{loss}\;(\%)=\frac{m_{s}-m_{d}}{m_{s}}\times 100$$
where the *m*_s_ is the mass of the swollen sample after immersing it in water for ~2 hours, and *m*_d_ is the mass of the sample after hydrolysis at the given time. The measurements were performed with three independent discs and the average results were reported.

Cytotoxicity tests: According to the reported method [35], mouse fibroblast cells (L929) were adopted as the test model to evaluate the cytotoxicity of possible substance that could leach from SMPEEU films. Before the cytotoxicity tests, SMPEEU films in the form of 10 ± 0.02 mm discs were rinsed with ethanol to eliminate the effects of impurity on the cells, and then were sterilized with UV radiation for 30 min on each side of the discs. The sterilized film discs were incubated in 2.0 mL culture medium (Dulbecco’s modified Eagle’s medium with 10% (*v*/*v*) fetal bovine serum) overnight at 37 ± 0.5 °C. The extract solution was filtrated with a filter membrane (0.22 μm pore size) to avoid the possible presence of solid particles released from the samples, and the filtrate was diluted with equivoluminal culture medium. L929 cells with a density of 5.0 × 10^4^ cells/mL were seeded into a 96-well tissue culture plate containing 100 μL of the respective film extracted dilutions, and wells which only contained the cells and culture medium served as controls. After incubation at 37 °C with 5% CO_2_ for 72 h, cell viability was measured with the thiazolyl blue tetrazolium bromide (MTT) assay method. The optical density (OD) was obtained with a multiwell microplate reader (Multiskan Mk3-Thermolabsystems, Thermo Fisher Scientific, Inc., Waltham, MA, USA) at 570 nm, and values relative to the control were reported.

Shape memory behaviors: The shape memory behaviors of SMPEEU films were evaluated with the “fold-deploy shape memory test” method [36,37] using rectangular strips with dimensions of 40 mm × 10 mm × 0.30 mm. First, the sample was immersed in a water bath with temperature of ~55 °C (25 °C above the *T*_g_) for 0.5 min, and then was folded in half under the aid of external force. Second, the folded sample was quickly quenched by immersion in an ice water bath (0 °C) under the existence of constant external force to fix the shape. A few minutes later (typically 1.5 min), the applied external force was removed and a marginal recovery occurred. The blending angle was recorded as *θ*_f_. Finally, the sample with the fixed shape was immersed in a water bath at body temperature (37.5 °C) and reheated to recover its original shape. The corresponding bending angle after recovery was recorded as *θ*_r_, and the time of the bending angle changing from *θ*_f_ to *θ*_r_ was recorded as the recovery time (*T*_r_). In order to quantitatively describe the shape memory properties of the films, the fixity ratio (*R*_f_) and recovery ratio (*R*_r_) were roughly defined as the following formulas (3) and (4), respectively:(3)$$R_{f}\;(\%)=\frac{\theta_{f}}{180}\times 100$$
(4)$$R_{r}\;(\%)=\frac{180-\theta_{r}}{180}\times 100$$

## 3. Results and Discussion

### 3.1. Characterization

^1^H NMR and FT-IR measurements were adopted to confirm the structure of the chain extender, prepolymer, and polyurethane. Representative ^1^H NMR and FT-IR spectra of HBH, PECL and SMPEEU are displayed in Figure 2 and Figure 3, respectively.

The chain extender of diurethane diisocyanate (HBH) was obtained through the reaction of eight-fold excess of HDI with BDO without using a catalyst. The data obtained from the ^1^H NMR spectrum (Figure 2a) were consistent with the results of our previous paper [33] and matched the chemical structure of HBH (Figure 1a), indicating no longer segments existing in the products. The peaks at 3319, 2262, 1680, and 1521 cm^−1^ in the FT-IR spectrum (Figure 3a) were attributed to the characteristic absorptions of –NH–, –NCO, amide I, and amide II, respectively. The FT-IR results provide additional support for the formation of the desired diurethane diisocyanate structure.

Typical signals appeared at δ 4.06, δ 2.29, and δ 3.64 ppm in the ^1^H NMR spectrum of PECL (Figure 2b), and were attributed to proton signals of ω-CH_2_, α-CH_2_ from [CL] units and repeat units of –O–CH_2_–CH_2_– ([EO]) from PEG segments, respectively [38]. According to the molecular weight of the PEG macroinitiator, the number of average repeat units of [CL] in the prepolymer could be obtained by calculating the peak integration at δ 4.06 and/or δ 2.29 with δ 3.64 ppm. The average repeat units of [CL] (${\bar{DP}}_{\mathrm{CL}}$) in the prepolymer and the molecular weight (*M*_NMR_) of the prepolymer were obtained according to Equations (5) and (6), respectively [39]:(5)$${\bar{DP}}_{CL}=\frac{M_{\mathrm{PEG}}}{44}\times \frac{S_{{\alpha ,\omega -\mathrm{CH}}_{2}}}{S_{\mathrm{EO}}}$$
(6)$$M_{NMR}=M_{PEG}+{\bar{DP}}_{CL}\times 114$$
where $S_{{\alpha ,\omega -\mathrm{CH}}_{2}}$ and $S_{\mathrm{EO}}$ are the peak integration of α,ω-CH_2_ proton signals from [CL] units and [EO] proton signals from PEG segments, and 44 and 144 are the molecular weight of [EO] and [CL] repeat units. The values of *M*_NMR_ matched with those of *M*_theo__._ calculated from the stoichiometry of CL/PEG (mol:mol) and *M*_n_ obtained from GPC (Table 1), indicating a complete ring-opening reaction and the absence of residual *ε*-CL in the PECL. In the FT-IR spectrum (Figure 3b), the stretching vibration of cyclic ester C=O (~1760 cm^−1^) disappeared completely, and another strong absorption band at ~1723 cm^−1^ appeared which should be attributed to the aliphatic ester C=O stretching vibration. In addition, the absorption bands at 3510 and 1092 cm^−1^ belonged to ether C–O–C stretching frequencies and terminal hydroxyl groups. This was further support that the ring-opening reaction indeed occurred.

SMPEEUs were prepared via chain extension of PECLs with HBH at 80 °C. The transesterification reaction, which had been reported to be severe when using diol as a chain extender at 80 °C [25], could be limited by using HBH diisocynantes as the chain extender, resulting in low *Ð*_M_ (Table 2), The chemical structures of the SMPEEUs were characterized by ^1^H NMR (Figure 2c) and FT-IR (Figure 3c). The proton signal of –CH_2_– next to –NCO (δ 3.33 ppm, ‘a’ in Figure 1a) and the absorption band of –NCO (~2262 cm^−1^ ) of HBH and the terminal –OH (~3319 cm^−1^) of PECL disappeared completely, alongside the appearance of the proton signals of –NH– in urethane groups and –CH_2_– next to urethane groups at δ 4.90 and 3.15 ppm, respectively (the signals appeared at δ 7.07 and 2.94 ppm of HBH in DMSO-d^6^, Figure 2a). In addition, the absorption peaks at 1093, 1168, 1535, 1675 and 1723 cm^−1^ in the FT-IR spectrum (Figure 3c) belonged to the stretching vibration of ether C–O–C, ester C–O–C, amide II, amide I, and aliphatic ester C=O, respectively. All the results represented a thorough chain extension reaction.

### 3.2. Thermal Transition

The typical DSC curves of the SMPEEU films with varying PEG content are presented in Figure 4 and the corresponding transition temperatures obtained from the thermograms are listed in Table 3. Two glass transition temperatures (*T*_g1_, *T*_g2_) were founded in their curves. The first glass transition points of *T*_g1_ −41.9 to −48.5 °C belonged to the soft domains, which was consistent with the result of polyurethanes-based poly(3-hydroxybutyrate) and PCL-PEG-PCL in a previous study [40]. Only one *T*_g_ appeared at a low temperature, indicating the high miscibility of PEG and PCL segments. The second glass transition points (*T*_g2_) appeared at higher temperature of ~32 °C, belonging to the well-defined hard domains. Because the polarity of urethane groups in hard segments is much higher than that of ester and ester groups in soft segments, they are hardly miscible, leading to microphase separation structure and two *T*_g_ for soft and hard domains [41]. The *T*_g1_ decreased slightly with the increment of PEG content in SMPEEU, which meant that the introduction of PEG could improve the flexibility of the SMPEEU films and act as a plasticizer [42]. Moreover, one broad endothermic peak (*T*_m_) at about 50~80 °C was observed, which should belong to the crystalline melting transition of PCL segments and hard domains. The Δ*H*_f_ decreased from 56.7 to 19.6 J/g (Table 3) with the PEG content increasing from 9.25 to 47.5 wt % in SMPEEU (Table 2), which signified that the PEG segments introduced to the main chain of the SMPEEU destroyed the crystallization of PCL and/or hard segments.

### 3.3. Crystallization Behaviors

The crystallization behaviors of the SMPEEU films were studied by means of XRD, and the results are presented in Figure 5. The SMPEEU-I film exhibited a broad diffraction zone with two clear diffraction peaks (2*θ*: 21.4°, 23.6°) in the scattering pattern, suggesting some degree of crystallinity. This was assigned to the crystalline soft domains (mainly PCL segments) and regular hard regions formed by the well-defined structure [43,44]. The intensity of the broad peaks decreased with the increasing PEG content in the SMPEEUs (SMPEEU-I~SMPEEU-IV), which indicated that the degree of crystallinity decreased gradually. It was supported that the introduction of PEG could affect the crystallinity of the SMPEEUs, and the results were consistent with DSC analysis.

### 3.4. Tensile Properties

Quintessential stress–strain curves of SMPEEU films with varying PEG content are displayed in Figure 6, and the tensile properties obtained from the curves are listed in Table 4. All the films first presented an elastic deformation, and then necking was observed clearly with a yield point, which indicated a smooth transition from elastic to plastic deformation. With PEG content increasing from 9.25 to 47.5 wt %, the strain at break increased from 483% to 956%, while the ultimate stress and initial modulus decreased from 23.1 to 9.0 MPa and 48 to 8.7 MPa, respectively (Table 4). The trend appears to be correlated with the PEG chain in the polymers which is much more flexible than a PCL chain of comparable molecular weight. The introduction of a flexible PEG chain hinders the crystallization of PCL soft segments and/or well-defined hard segments, and weakens the average crystallite size as shown by XRD analysis [45]. Consequently, the tensile properties can be tailored by adjustment of the PEG content to meet the requirements of medical materials.

### 3.5. Bulk and Surface Hydrophilicity

The bulk and surface hydrophilicity, which can be reflected directly by the measurements of water absorption and water surface contact angle, are very significant for biomaterials because the wettability can affect the hydrolytic degradation rate and the interaction of cells with the materials [46]. The water contact angle and water absorption of SMPEEU films with varying PEG contents are shown in Figure 7 and Figure 8, respectively. With the increment of PEG content in SMPEEUs, the water contact angle gradually decreased from 72.8° to 41.7°, indicating an improving surface hydrophilicity. The decrease in water contact angles should be ascribed to the excellent hydrophilicity of PEG chains exposed on the surfaces, and the better surface hydrophilicity means higher biocompatibility, such as reduced tissue adhesion [42]. The water absorption of the films occurred with a rapid water penetration at early immersion and reached the equilibrium after ~90 min (Figure 8), displaying a Fickian diffusion behavior [47,48]. The equilibrium water absorption increased sharply from 3.1 to 23.6 wt % with the increasing amount of PEG incorporated into the polymer structure, which is also attributed to the unique hydration effect of the PEG. The intermolecular forces between water and the PEG chains caused the water to actively penetrate into the films. From the results, the bulk and surface hydrophilic ability—important roles in the hydrolytic stability—are mainly influenced by the hydrophilicity of the components [49]. Therefore, it is deduced that the degradation rate of the SMPEEU films can be affected by the content of hydrophilic PEG.

### 3.6. In Vitro Hydrolytic Degradation

In vitro hydrolytic degradation tests of the SMPEEU films were carried out at 37 °C in PBS solution with a pH value of 7.4. The time dependence of the percentage weight loss is shown in Figure 9. The degradation process included two stages: the samples presented a slight weight loss of less than 10 wt % in the first two weeks, followed by a sharp increase in the weight loss rate up to the end of the test (degradation criterion is the film becoming fragments). The swelling and hydration of the films are contained in the incipient stage, which leads to a slow weight loss rate in appearance. The substantial weight loss in the second stage should be due to the loss of water-soluble molecules formed during the hydrolysis of the ester groups. Obviously, introduction of hydrophilic PEG could enhance the degradation rate of the polymers, and the time of the SMPEEU-I, -II, -III, and -IV films becoming fragments was 12, 9, 6, and 4 weeks, respectively. The trend can be explained by two factors: PEG content and crystallinity. As in the description of the measurement of water absorption, high hydrophilic PEG content can bind more water molecules. The introduction of PEG can slightly destroy the order of the polymer structure and decrease the crystallinity of the polymer, as verified in XRD, which allows the water molecules to easily pass through the film. Thus, the ester groups approached the water molecules easily, generating an enhanced hydrolytic degradation rate.

### 3.7. Shape Memory Properties

The shape memory properties are the most significant characteristics by which to evaluate the quality of SMPs. The purpose of this paper is to offer a new shape-memory biomaterial for medical device applications, so the recovery process is carried out at body temperature—37.5 °C.

The shape memory properties, including the fixity ratio (*R*_f_), recovery ratio (*R*_r_), and recovery time (*T*_r_), are quantitatively measured to assess the shape memory behaviors of the developed SMPEEU films, and the results are displayed in Table 5. In the process for the “fold-deploy shape memory test”, the fixed angle *θ*_f_ was almost equal to 180° (less than 1°), and *R*_f_ is more than 99.5%, so that the films had high *R*_f_ and could be fixed to the desired shape completely. After the shape-fixed films were immerged in a 37.5 °C water bath, all the samples needed only several seconds to recover their original shape, indicative of an excellent shape recovery performance. As previous papers have reported [20,21,50,51], the hydrogen bonds play an important role in the structure change in the deformation and shape memory processes, so it can be assumed that the excellent shape memory properties are related to the denser hydrogen bonds in the SMPEEU structure. Four urethane groups are contained in each hard segment; therefore, multiple hydrogen bonds can exist not only among the hard segments but also between the hard and soft segments. The influence mechanism of denser hydrogen bonds on the shape memory performance needs further research. In addition, the crystalline PCL chain segments in the soft phase contributed to the shape memory performance [52,53]. The SMPEEU with a PEG content of 23.4 wt % (SMPEEU-III) manifested more excellent recovery properties with an *R*_r_ of 99.8% and *T*_r_ of 3.9 s, meaning that introduction of flexible PEG chain into the SMPEEU structure improved the recoverability at body temperature and the optimum PEG content in SMPEEUs was approximately 23.4 wt %. The representative recovery process of the an SMPEEU-III film is visualized in Figure 10. In order to more directly exhibit the shape memory performances, the SMPEEU-III film was designed as “A” shape and dyed, and the shape fixity and recovery procedure was demonstrated in Appendix A in the ESI^†^. The deformed shape of the random clew needs only 3.5 s to recover its original “A” shape.

Repeatability of shape memory recovery, which can characterize the fatigue resistance of materials, is an important performance parameter for SMPs. The influence of repeated fold-deploy cycles on the recovery properties of *R*_r_ and *T*_r_ was studied, and the results are shown in Figure 11. The SMPEEU-III film could almost recover its original shape rapidly over four cycles. The slight decrease of *T*_r_ may be due to the adjustment of multiple hydrogen bonds existing in the SMPEEU structures. The friction among molecules was reduced with frequent bending, and thereafter, the molecules became more submissive, resulting in a reduced *T*_r_ [36,54]. During the subsequent cycles, the *R*_r_ and recovery rate decreased dramatically, which could be caused by material fatigue, demonstrating that the multiple hydrogen bonds were partly destroyed so that the material gradually lost its shape memory property.

### 3.8. Cytotoxicity Test

The first step to evaluate the biocompatibility of biomedical materials is the cytotoxicity test. L929 cells were cultured with SMPEEU film extracts to measure whether the film extracts were toxic to cells. From the average OD values calculated with Equation (7), the relative growth rate (RGR) of the test biomaterial can be obtained. An RGR ≥ 75% implies low or no cytotoxicity, and the material can be utilized in medical applications [35]. The results of an MTT assay of L929 cells in SMPEEU film extracts after 72 h are presented in Figure 12. Although a slight reduction of RGR was observed compared with the control, all the RGRs were larger than 75%, indicating that the SMPEEU films had low toxicity to cells in vitro and met the requirement of medical materials. The RGR increased gradually with the increase of PEG content, which can be attributed to the excellent cytocompatibility of PEG.

(7)$$\mathrm{RGR}\;(\%)=\frac{\mathrm{OD}\;\mathrm{value}\;\mathrm{of}\;\mathrm{sample}\;\mathrm{suspension}}{\mathrm{OD}\;\mathrm{value}\;\mathrm{of}\;\mathrm{negative}\;\mathrm{control}\;\mathrm{suspension}}\times 100$$

## 4. Conclusions

In this paper, a new class of degradable SMPEEUs with well-defined hard segments was prepared by one-step chain extension. The chemical structures of PECLs and SMPEEUs were confirmed by ^1^H NMR, FT-IR, and GPC. The effect of PEG content on the physicochemical properties, including the thermal transition, crystallization behavior, bulk and surface hydrophilicity, tensile properties, and in vitro degradability, was extensively studied. DSC and XRD research demonstrated that the SMPEEUs had a microphase-separated structure, and the introduction of PEG segments could lead to a low transition temperature and decrease the crystallinity of the polymers. The surface and bulk hydrophilicity of SMPEEU films were found to be closely related to the hydrophilic PEG content. With the PEG content increasing from 9.25 to 47.5 wt %, the strain at break increased from 483 to 956%, and the ultimate stress decreased from 23.1 to 9.0 MPa. Hydrolytic degradation in vitro studies indicated that the time of the SMPEEU films becoming fragments was 4–12 weeks, and the introduction of PEG could facilitate the degradation rate of the films. The shape memory properties were evaluated by the fold-deploy test. The SMPEEU-film with a PEG content of 23.4 wt % displayed excellent recovery properties with a recovery ratio of 99.8% and recovery time of 3.9 s at body temperature, and the film could almost recover its original shape rapidly after four fold-deploy cycles. In addition, a cytotoxicity test of the film extracts was conducted using L929 cell, and the RGR was larger than 75% after incubation for 72 h, indicating good cytocompatibility in vitro. The SMPEEUs possess not only satisfactory tensile properties, biodegradability, nontoxic degradation products, and good cytocompatibility, but also excellent recovery properties at body temperature, suggesting their potential for application as biomedical devices.

## Figures and Tables

**Figure 1 polymers-11-01002-f001:**
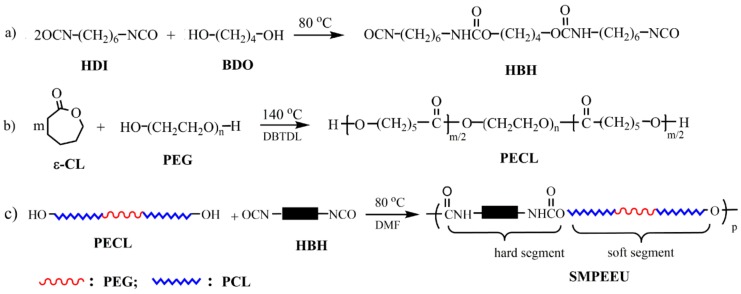
Synthetic pathways of (**a**) diurethane diisocyanate (HBH), (**b**) PECL, and (**c**) SMPEEU.

**Figure 2 polymers-11-01002-f002:**
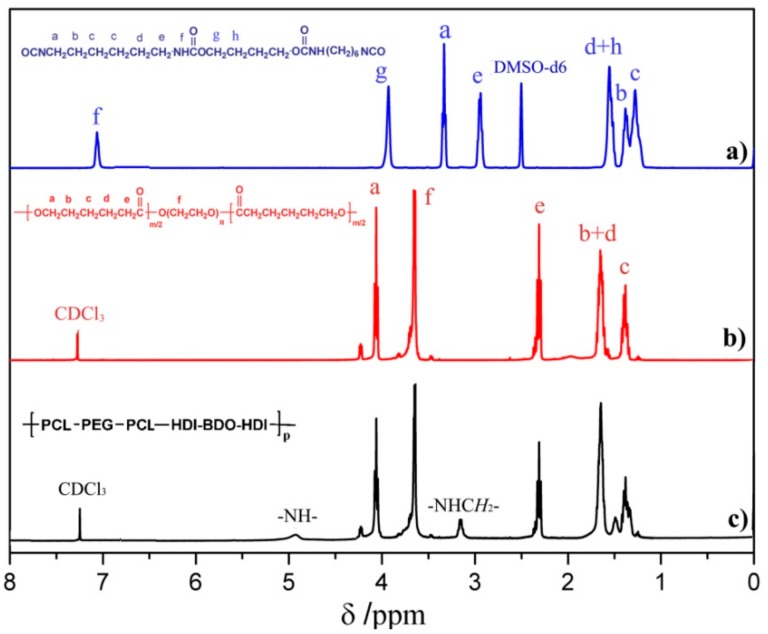
Representative ^1^H NMR spectra of (**a**) HBH, (**b**) PECL-III, and (**c**) SMPEEU-III.

**Figure 3 polymers-11-01002-f003:**
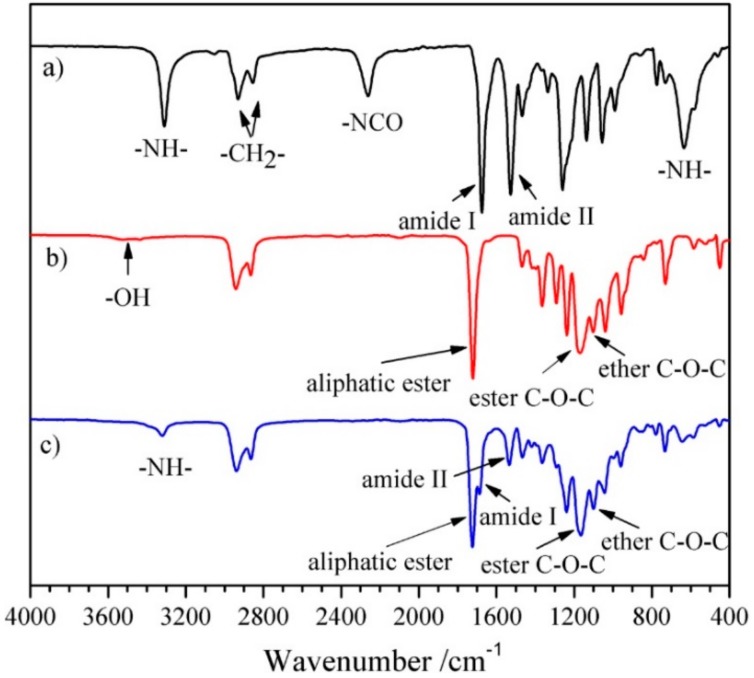
Representative FT-IR spectra of (**a**) HBH, (**b**) PECL-III, and (**c**) SMPEEU-III.

**Figure 4 polymers-11-01002-f004:**
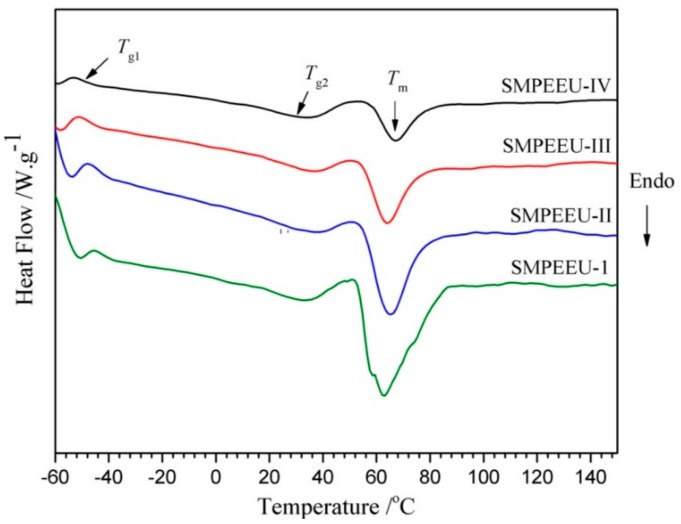
The typical differential scanning colorimetry (DSC) curves of the SMPEEU films in the second heating cycle.

**Figure 5 polymers-11-01002-f005:**
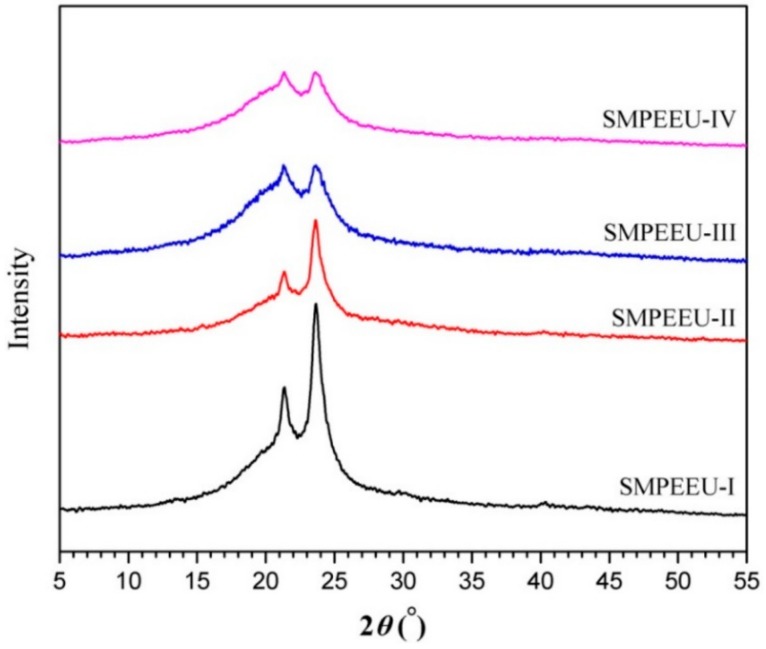
X-ray diffraction (XRD) patterns of SMPEEU films.

**Figure 6 polymers-11-01002-f006:**
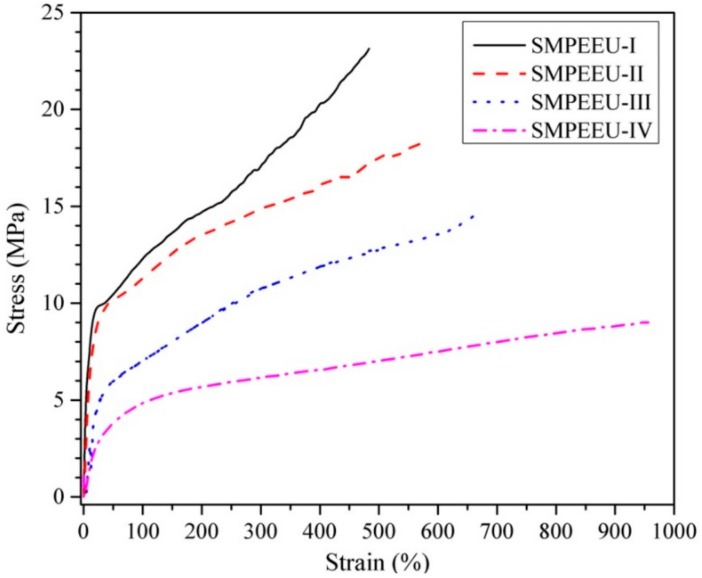
Stress-strain behaviors of SMPEEU films.

**Figure 7 polymers-11-01002-f007:**
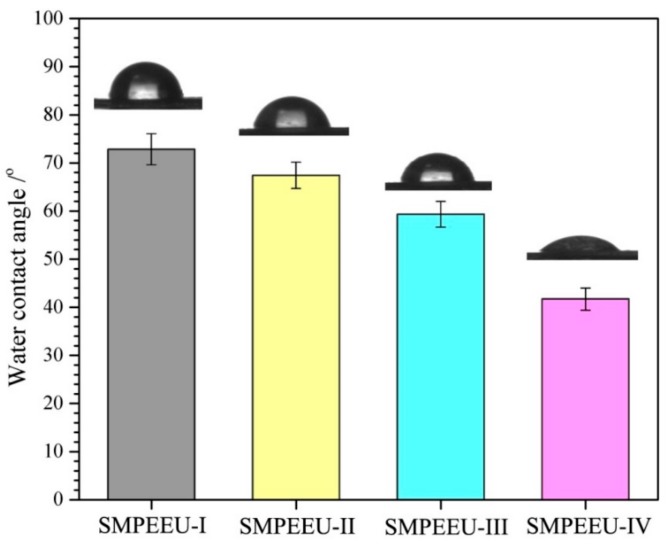
Water surface contact angle of SMPEEU films with varying PEG contents. (Error bars represent standard error of mean; *n* = 6).

**Figure 8 polymers-11-01002-f008:**
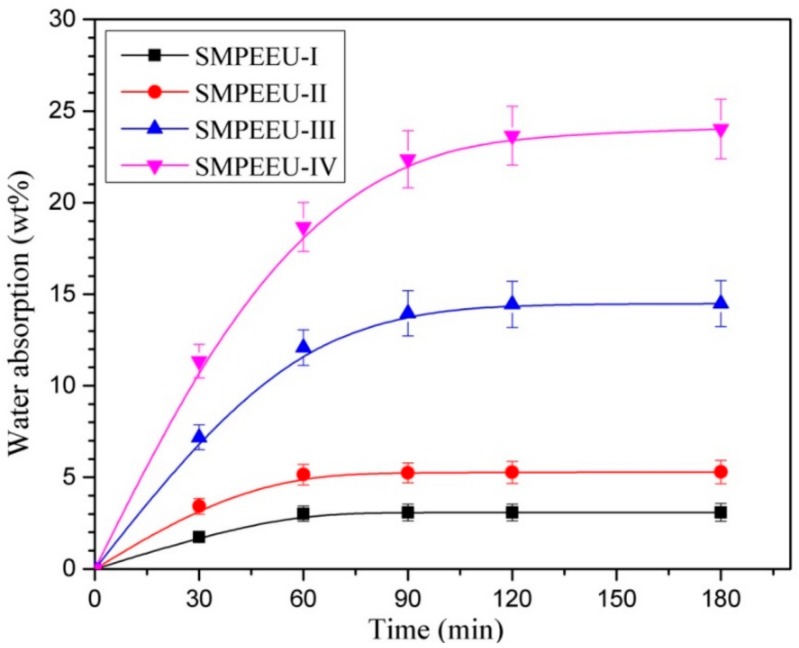
Water absorption of SMPEEU films with varying PEG contents. (Error bars represent standard error of mean; *n* = 3).

**Figure 9 polymers-11-01002-f009:**
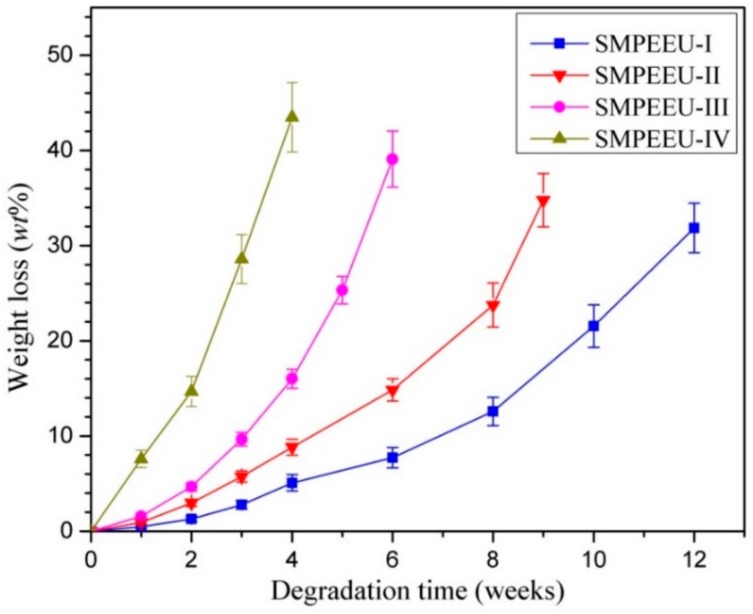
Degradation behaviors of SMPEEU films with varying PEG content. (Error bars represent standard error of mean; *n* = 5).

**Figure 10 polymers-11-01002-f010:**
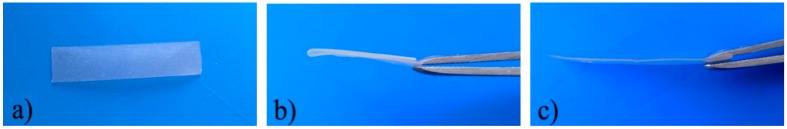
Representative shape memory process of an SMPEEU-III film. (**a**) original shape; (**b**) fixed shape; (**c**) recovered shape.

**Figure 11 polymers-11-01002-f011:**
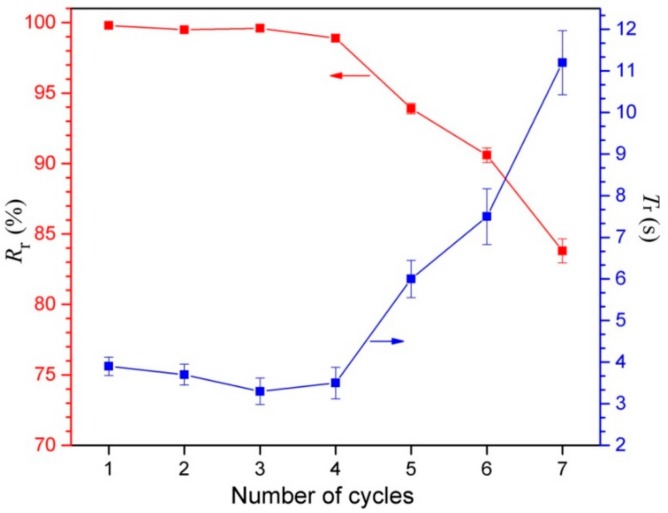
Shape memory properties of an SMPEEU-III film after several fold-deploy cycles. (Error bars represent standard error of mean; *n* = 3).

**Figure 12 polymers-11-01002-f012:**
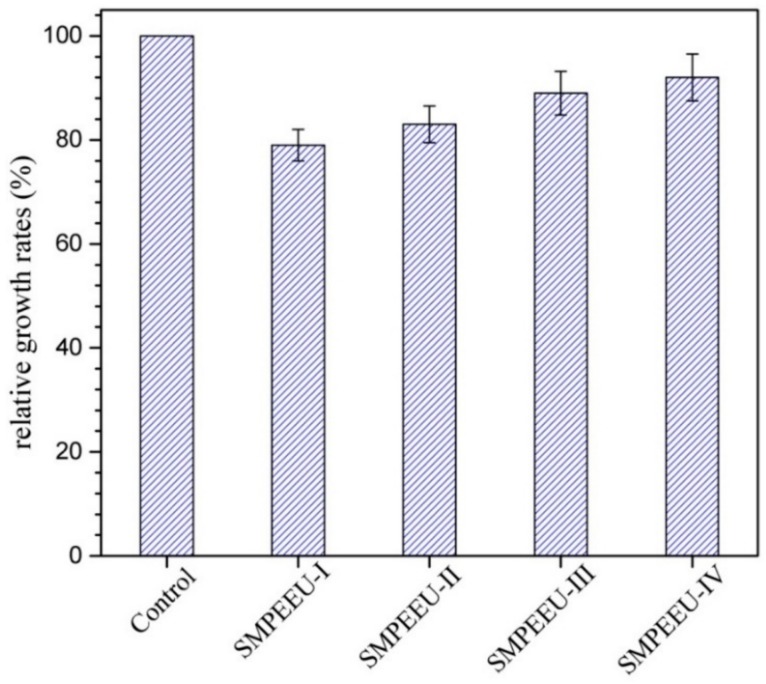
MTT assay of L929 cells on SMPEEU film extracts. (Error bars represent standard error of mean; *n* = 3).

**Table 1 polymers-11-01002-t001:** Molecular weight and feed ratios of the poly(ether-ester) (PECL).

PECLs	*ε*-CL/g	PEG/g				*M* _theo_	*M* _n_	*Ð* _M_	*M* _NMR_
		−400	−600	−1000	−2000				
-I	36	4.0	-	-	-	4000	3890	1.11	3990
-II	34	-	6.0	-	-	4000	3920	1.08	4020
-III	30	-	-	10.0	-	4000	3880	1.10	3970
-IV	20	-	-	-	20.0	4000	3940	1.12	4040

Note: *M*_theo._, theoretical molecular weight calculated from the molar ratio of CL/PEG; *M*_n_ and *Ð*_M_, number average molecular weight and molecular weight dispersity obtained from GPC; *M*_NMR_, molecular weight calculated from the peak integration in ^1^H NMR.

**Table 2 polymers-11-01002-t002:** Molecular weights and feed ratios of the shape-memory poly(ether-ester-urethane)s (SMPEEUs).

SMPEEUs	PECL/g				HBH/g	PEG Content/wt %	*M*_n_ (kDa)	*Ð* _M_
	-I	-II	-III	-IV				
-I	19.5	-	-	-	2.17	9.25	118	1.38
-II	-	19.6	-	-	2.17	13.8	113	1.41
-III	-	-	19.4		2.17	23.2	115	1.43
-IV	-	-	-	19.7	2.17	45.7	108	1.37

Note: PEG content, PEG content in SMPEEUs; *M*_n_ and *Ð*_M_, number average molecular weight and molecular weight dispersity obtained by gel permeation chromatography (GPC).

**Table 3 polymers-11-01002-t003:** The transition temperatures of SMPEEU films.

SMPEEUs	-I	-II	-III	-IV
*T*_g1_ (°C)	−41.9	−43.7	−46.1	−48.5
*T*_g2_ (°C)	30.4	32.8	33.4	31.6
*T*_m_ (°C)	52–81	53–80	53–77	55–78
Δ*H*_f_ (J/g)	56.7	43.8	29.3	19.6

**Table 4 polymers-11-01002-t004:** Tensile properties of SMPEEU films.

Films	Strain at Break (%)	Ultimate Stress (MPa)	Yield Stress (MPa)	Yield Strain (%)	Initial Modulus (MPa)
SMPEEU-I	483 ± 12	23.1 ± 1.7	9.7 ± 0.76	20.2 ± 1.2	48.0
SMPEEU-II	589 ± 15	18.5 ± 1.5	9.5 ± 0.69	24.0 ± 1.4	39.6
SMPEEU-III	675 ± 18	14.8 ± 1.3	5.8 ± 0.52	34.1 ± 1.6	17.0
SMPEEU-IV	956 ± 23	9.0 ± 0.9	4.7 ± 0.27	54.1 ± 1.9	8.7

Note: “±” represents standard error of mean; *n* = 3.

**Table 5 polymers-11-01002-t005:** Shape memory properties of SMPEEU films.

Films	Thickness (mm)	*R*_f_ (%)	*R*_r_ (%)	*T*_r_ (s)
SMPEEU-I	0.30	99.5 ± 0.02	81.1 ± 1.4	12.2 ± 0.5
SMPEEU-II	0.31	99.6 ± 0.02	90.7 ± 1.2	7.5 ± 0.4
SMPEEU-III	0.30	99.7 ± 0.01	99.8 ± 0.06	3.9 ± 0.2
SMPEEU-IV	0.29	99.8 ± 0.01	92.8 ± 0.7	8.4 ± 0.5

Note: “±” represents standard error of mean; *n* = 3.

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
