# Peer review of "Degradable Poly(ether-ester-urethane)s Based on Well-Defined Aliphatic Diurethane Diisocyanate with Excellent Shape Recovery Properties at Body Temperature for Biomedical Application"

_polymers, 2019, doi:10.3390/polym11061002_

Round 1

Reviewer 1 Report

Overall, this is a sound, if methodical manuscript with minor issues with grammar and spelling. There are also instances of abbreviations not being defined (particularly in tables) or defined inaccurately (for example, Mn is a Number average molecular weight). Similarly some of the bond angles in the schemes could be better and made neater. Overall though, the manuscript really suffers from a lack of discussion of the results. This is especially true of the decrease in shape fixity results that are described as a disruption in hydrogen bonding with no underlying evidence or cause proposed. In my opinion, these issues need to be addressed before publication.

Author Response

Please see the response for review 1 in the attached file.

Reviewer 2 Report

See the attached document.

Author Response

Please see the response for reviewer 2 in the attached file.

Reviewer 3 Report

In this manuscript, the authors have reported the preparation and investigation of biodegradable and biocompatible poly(ether-ester-urethane)s based on  aliphatic diurethane diisocyanate. The topic of this paper is interesting and important. The manuscript is well-written and organized and it is suitable for publication in Polymers after some minor revisions outlined below.

-          The utility of PCL in the shape memory PU is not mentioned and discussed.

-          In Table 1 the MWD-s were most probably determined from GPC and not by theoretical calculations as indicated. Please check The Table 1!

-          Figure 2. is of poor quality, it should be changed!

-          In line 239 the authors use the triblock expression (which is hold for copolymers) for the HDI terminated BDO. Please correct it!

-          Equations 4 and 5 should be moved from the R&D into the Experimental part.

-          In line 283 ”wll-defined” should read as ”well-defined”.

-          In the heading of Table 3. ”-I –II –II –III” should read as ”-I –II –III –IV”.

Author Response

Please see the response for reviewer 3 in the attached file.

Reviewer 4 Report

This paper presented an interesting study of biodegradable shape memory polymers for potential biomedical applications. The authors first reviewed the state of the art for shape memory polymers. Then, detailed experimental study procedures were presented. Adequate results were well-organized and easy to follow in the current paper. The reviewer suggests publishing this paper after a minor revision. Please see detailed comments listed below.

The journal requests all abstract to have 200 words maximum. The current abstract has nearly 400 words. Please revise the abstract and provide a concise abstract in the revised paper. 

A significant amount of work related to biocompatible shape memory polymers and foams have been published recently. Please consider refer to the following papers in the revised draft.

Kunkel et al. Journal of the Mechanical Behavior of Biomedical Materials, 88: 422-430, 2018.

Wang et al. Polymers, 11(4): 631, 2019.

Wang et al. Materials Letters, 250, 38-41, 2019.

Please clarify the minimum temperature required to trigger the shape recovery process.

Author Response

Please see the response for reviewer 4 in the attached file.

Round 2

Reviewer 1 Report

The authors have made the changes suggested by the reviewers.

Reviewer 2 Report

The proper IUPAC notation for molecular weight dispersity is ÐNOT MwÐ.